# **CLIMATHUNDERR: Experimental database of buoyancy-driven downbursts**

Federico Canepa<sup>1</sup>, Anthony Guibert<sup>2</sup>, Andi Xhelaj<sup>1</sup>, Josip Žužul<sup>1</sup>, Djordje Romanic<sup>3</sup>, Alessio Ricci<sup>4</sup>, Horia Hangan<sup>5</sup>, Jean-Paul Bouchet<sup>2</sup>, Philippe Delpech<sup>2</sup>, Olivier Flamand<sup>2</sup>, Massimiliano Burlando<sup>1</sup>

- <sup>1</sup>Department of Civil, Chemical and Environmental Engineering (DICCA), University of Genoa, 16145, Genoa, Italy
   <sup>2</sup>Centre Scientifique et Technique du Bâtiment (CSTB), 44300, Nantes, France
   <sup>3</sup>Department of Atmospheric and Oceanic Sciences, McGill University, Montreal, Quebec H3A 0E2, Canada
   <sup>4</sup>Department of Science, Technology and Society, University School for Advanced Studies IUSS Pavia, 27100, Pavia, Italy
   <sup>5</sup>Faculty of Engineering and Applied Science, Ontario Tech University, Oshawa, Ontario L1G 0C5, Canada
- Correspondence to: Federico Canepa (federico.canepa@unige.it)

Abstract. Thunderstorm downbursts are windstorms due to intense negatively-buoyant flows produced beneath cumulonimbus clouds. Their study has recently attracted significant scientific and media attention due to the current and projected impacts of climate change. During their vertical descent phase (i.e., the downdraft), followed by a horizontal outflow, downbursts can cause severe damage to both natural ecosystems and built environments. Warm, humid air is lifted

- upward through natural or forced convective mechanisms, where it condenses into a cumulonimbus cloud. Inside the cloud, the air parcels—now colder and denser than the surrounding environment—sink due to buoyancy. Thermal and dynamic instabilities between the cold air jet and the environment generate a symmetrical vortex, known as the primary vortex (PV), which drives both the downdraft and the subsequent horizontal outflow at the surface. This vortex flow structure can have devastating effects on the ground.
- Building on these insights, a series of experiments was recently conducted as part of the CLIMATHUNDERR project— CLIMAtic Investigation of THUNDERstorm Winds—funded by the European Union through the European Research Infrastructures for European Synergies (ERIES) project. For the first time, the buoyancy effects that drive downdraft winds to the surface were reproduced at large fluid-dynamics geometric scales at the Jules Verne Climatic Wind Tunnel—Thermal Unit SC2 at the Centre Scientifique et Technique du Bâtiment (CSTB) in Nantes, France. This experimental campaign aimed
- to further explore thunderstorm wind phenomena, building on earlier research studies conducted at the WindEEE Dome in Canada under the European Research Council (ERC) Advanced Grant project THUNDERR. CLIMATHUNDERR extends this previous research by emphasizing thermal effects, which are key drivers in these wind events. In the experiments, downbursts were recreated using an upper plenum that simulates the thunderstorm cloud, innovatively combining two widely applied techniques: impinging jet and gravity current. Thermal effects were reproduced by controlling the temperature
- differential between the upper plenum and the air in the testing chamber. A mechanical piston controlled the outgoing flow velocity at the nozzle exit, simulating the contribution of a simple mechanical impinging jet. Benchmark experiments were

performed with only the mechanical impinging jet, allowing the quantification of thermal effects at the interface between the jet and the calm surrounding air.

The experimentally generated downburst-like flows were then tested against a scaled orography model of the Polcevera Valley in Genoa, Italy, to examine how it influences the dynamics and structure of the downburst vortices.

- Velocity measurements were performed using Particle Image Velocimetry (PIV), enabling a detailed reconstruction of the 2D vector flow field without the limitations of traditional anemometric instruments like multi-hole pressure probes, which struggle with low-velocity (i.e.,  $

- Addressing the complexity of downbursts requires either controlled physical simulations in specialized laboratories or 65 numerical modeling. While advancements in computational power have significantly improved the ability to replicate the three-dimensional and transient nature of these phenomena (Žužul et al., 2024), computational fluid dynamics (CFD) models still struggle to accurately capture the turbulent component and instabilities—combined to thermal effects—of the flow at reasonable computational costs. Experimental methods are well-established but only a few laboratories worldwide are capable of reproducing downbursts at the relevant Reynolds numbers for civil and wind engineering applications. Currently, 70 no wind simulator can fully reproduce the thermomechanical mechanisms responsible for the formation and evolution of
- downburst winds at large geometric and kinematic scales. A downburst onsets when dense, cold air from a thunderstorm cloud descends as a vertical downdraft. Upon reaching the ground, the downdraft's momentum shifts from vertical to horizontal, creating intense radial winds near the surface, the so-
- 75 current (GC) method (Simpson, 1969; Charba, 1974; Lundgren et al., 1992; Yao and Lundgren, 1996), where the downburst is driven by the buoyant force arising from the density difference between the heavier downdraft and the lighter surrounding fluid; and (ii) the impinging jet (IJ) method (Brady and Ludwig, 1963; Canepa et al., 2022a; Didden and Ho, 1985; Gutmark et al., 1978; McConville et al., 2009; Romanic et al., 2019; Sengupta and Sarkar, 2008; Xu and Hangan, 2008), where the downburst is mechanically initiated by forcing air into the environment using air fans, generating a jet of similar temperature

called downburst outflow. Two experimental approaches are typically used to simulate downburst-like winds: (i) the gravity

- 80 and density. The GC method is more accurate in replicating the real-world conditions of a downburst, including the horizontal pressure gradients caused by air density and temperature differences. However, these experiments are often performed with fluids other than air to match non-dimensional parameters, such as the Richardson number, resulting in low velocities and small-scale experiments that are less relevant for practical engineering applications. In contrast, the IJ method does not fully replicate the thermodynamic processes of downbursts but it can reproduce the spatial and temporal evolution
- 85 of the near-ground flow field with higher velocities, making it more suitable for structural and environmental investigations. As a result, the IJ approach has been more widely adopted in recent research and is increasingly being integrated into design recommendations.

However, this raises an important question: Does the IJ method accurately simulate downburst flow at the ground, or do buoyancy effects significantly influence the downburst's geometric and dynamic evolution? In nature, the cold and dense air

- 90 from a thunderstorm cloud falls due to thermal contrast with the surrounding atmosphere (as seen in GC experiments) creating a downdraft. Despite lateral entrainment of warmer air into the jet, weather stations typically record temperature drops of up to 10°C during the passage of a downburst outflow (Choi, 2004; Choi and Hidayat, 2002; Huang et al., 2019). The greater is the available energy at the ground in the form of warm and humid air—usually measured by the Convective Available Potential Energy (CAPE) parameter—the higher is the potential for intense updrafts and formation of intense
- 95 cumulus clouds that may produce violent thunderstorms. The Downdraft Convective Available Potential Energy (DCAPE) considers the thermodynamic characteristics of the mid and lower parts of the atmosphere, often below the storm cloud base, to assess the energy that an air parcel might gain as it descends toward the surface. High DCAPE values indicate the

potential for stronger downdrafts approaching the ground, resulting in a more vigorous near-ground outflow. This raises further questions about how gravity currents and the temperature difference between the downdraft and surrounding air may affect the geometric characteristics and size of vortex structures in the downburst system. When applied at larger geometric

- 100 affect the geometric characteristics and size of vortex structures in the downburst system. When applied at larger geometric scales and using air with varying thermal properties, the GC technique could ultimately provide answers to these questions. To address these aspects, the CLIMATHUNDERR project (CLIMAtic Investigation of THUNDERstorm Winds) was initiated at the Jules Verne Climatic Wind Tunnel (JVCWT) Thermal Unit SC2 at the Centre Scientifique et Technique du Bâtiment (CSTB) in Nantes, France. Funded by the European Union's European Research Infrastructures for European
- 105 Synergies (ERIES) project, this project builds on the experimental campaigns (Canepa et al., 2022b, 2022c, 2022a, 2023, 2024), conducted in recent years at the WindEEE Dome at Western University in Canada, under the ERC project THUNDERR Detection, simulation, modelling and loading of thunderstorm outflows to design wind-safer and cost-efficient structures (Solari et al., 2020). The CLIMATHUNDERR acronym is intentionally inspired by the previous THUNDERR project, reflecting continuity while expanding its focus to include the thermal effects on thunderstorm winds.
- 110 In this project, varying thermal contrasts between the jet and the ambient air were tested to investigate the buoyancy effects on the velocity, dynamics, and geometric features of the downburst at the ground level. Large-scale Particle Image Velocimetry (LS-PIV) was employed to capture variations in the flow field, and thermocouples were used to monitor temperature profiles. To the authors' knowledge, this experimental campaign is unique, as the GC approach has not previously been applied at such large geometric scales and using only air as fluid.
- Additionally, the reproduced downburst flows were tested over a 1:2000 scale model of the Polcevera Valley in the municipality of Genoa, Italy, to assess the influence of local orography on the flow dynamics and vortex structures. Meteorological and anemometric measurements (Solari et al., 2012; Repetto et al., 2018; Burlando et al., 2018, 2020; Canepa et al., 2020, 2024; De Gaetano et al., 2014) provide compelling evidence that this region is highly susceptible to the formation of thunderstorms. This vulnerability stems from the proximity of the Mediterranean Sea, serving as a source of
- 120 warm and humid air essential for the initiation of air updrafts. Additionally, the Ligurian Apennines mountain range plays a crucial role by ushering in cold air. This combination creates intense convective conditions that often lead to downbursts approaching Genoa from the south-southwest. This setup was replicated in the JVCWT experiments. All data from the project are available to the public via the Zenodo repository and can be re-used under Creative Commons
- license CC0 for metadata and CC-BY for data (Canepa et al., 2025).
  The paper is organized as follows: Section 2 provides an overview of the CLIMATHUNDERR project, followed by a detailed description of the experimental setup, specimens, and instrumentation specifications in Section 3. Section 4 offers guidance on using the published dataset, while Section 5 provides a preview of its content. Finally, Section 6 closes the
  - paper with conclusions and perspectives.

# **2 CLIMATHUNDERR project**

- 130 The CLIMATHUNDERR project—CLIMAtic Investigation of THUNDERstorm Winds—was selected by the ERIES' evaluation panel to conduct experimental research at the JVCWT facility in the CSTB laboratory, Nantes, France. The project is funded by the European Commission's Horizon 2021 program. The User Group (UG) consists of researchers from leading universities in Italy and Canada, bringing together a multidisciplinary expertise on thunderstorm winds.
- UG members come from a wide range of disciplines, including atmospheric physics, meteorology, experimental and 135 numerical fluid dynamics, as well as civil and mechanical engineering. Collectively, they bring extensive expertise in studying thunderstorm winds from multiple perspectives. This includes full-scale field campaigns using anemometric and LiDAR profiler instruments (Burlando et al., 2017, 2018, 2020; Canepa et al., 2020, 2024b; Romanic et al., 2020c; Romanic, 2021; Zhang et al., 2018), experimental studies utilizing anemometric and PIV measurements (Canepa et al., 2023, 2022a, 2022b, 2022c; Canepa et al., 2024; Hangan et al., 2019; Junayed et al., 2019; Romanic et al., 2020a, 2020c, 2019; Romanic
- 140 and Hangan, 2020; Xu and Hangan, 2008) at the WindEEE Dome, one of the largest wind simulators capable of reproducing large-scale, non-stationary extreme wind events (Hangan et al., 2017). Additionally, numerical modeling techniques have been employed in various studies (Kim and Hangan, 2007; Žužul et al., 2023, 2024). These combined experimental and numerical approaches are being synthesized to develop a state-of-the-art analytical model for thunderstorm winds (Xhelaj et al., 2020; Xhelaj and Burlando, 2022, 2024). The model will be further enhanced by incorporating results from the current
- experiments to account for the thermal effects on downburst wind evolution. The implementation of the CLIMATHUNDERR experimental campaign was made possible through the crucial contributions of the technical staff at CSTB. The design of the experimental setup involved close collaboration between the UG and the CSTB technical staff, resulting in a highly complex, innovative, and unique experimental apparatus, specifically crafted to meet the project's objectives.

#### 150 **3** Experimental setup

145

The experiments were conducted in June 2024 at the JVCWT – Thermal Unit SC2, located at CSTB in Nantes, France. This section provides a detailed description of the thermal-wind simulator, the specimen and measuring techniques involved, as well as the overall test plan and operational setup.

#### **3.1 Facility**

The JVCWT, built in the 1990s, consists of two concentric wind tunnels: the outer ring is called "dynamic circuit" and the 155 inner one is called "thermal circuit". These wind tunnels allow comprehensive and full-scale aero-climatic simulations. The thermal circuit, where all data in this study was collected, can replicate a wide range of real-world climatic conditions, including rain, snow, frost, and solar radiation.

Figure 1 shows a schematic of the thermal circuit wind tunnel. The test section measures 10 m in width (W), 7 m in height (H), and 25 m in length (L). At the downstream end, a sudden contraction leads into a  $180^{\circ}$  turn. Upstream of the fan, the

- (H), and 25 m in length (L). At the downstream end, a sudden contraction leads into a 180° turn. Upstream of the fan, the cross-section transitions from rectangular to circular, with a diameter of 6.2 meters to guide the airflow into the fan. The variable-speed axial fan, with a power of 1100 kW, can generate steady wind speeds between 1 and 40 m/s. A heat exchanger is positioned downstream of the fan blades, while the adjustable nozzle creates a contraction from the cross-section after the second 180° turn (6 x 9 m<sup>2</sup>) to an area ranging from 6 x 5 m<sup>2</sup> to 6 x 3 m<sup>2</sup>. This contraction is controlled by
- lowering the ceiling height over a distance of 3 to 5 meters, allowing the wind flow to discharge into the larger test section through the nozzle exit.

This climatic wind tunnel differs from a typical boundary layer wind tunnel in several key aspects. Notably, the shorter fetch, presence of flow disturbances and separations, and a longitudinal pressure gradient all contribute to challenges in generating a stable atmospheric boundary layer (ABL) within the testing chamber.

Figure 1. Schematic of the thermal circuit wind tunnel with relevant dimensions of the testing chamber. Yellow arrow shows the upstream/downstream orientations.

# 170 3.2 Specimen and geometry

The primary technical challenge of the experimental setup involved creating a downdraft-like jet from scratch. This required a mechanism capable of producing a vertical, top-down air jet within the testing chamber. In recent years, various wind

simulators have been developed to generate vertical air flows, either by blowing or sucking air, to simulate top-down or bottom-up jets that closely mimic natural downbursts or tornadoes, respectively (Haan et al., 2008; McConville et al., 2009;

- Hangan et al., 2017; Li et al., 2024). For this setup, an upper plenum with dimensions of  $2 \times 2 \times 2$  m<sup>3</sup> was constructed, hereafter referred to as the impinging-jet plenum (IJP). The IJP's walls were made of 15-mm thick plywood lined with insulation material to minimize thermal losses and prevent air stratification, in the view of ensuring a stable temperature differential ( $\Delta T$ ) between the plenum and the testing chamber. Figure 2a,b show the IJP installed in the testing chamber during the tests. A 7.3 kW Coolmobile 25 air conditioning unit, manufactured by Thermobile Industries B.V., was installed
- outside one of the IJP's side walls and connected via three tubes to cool the internal air to a minimum temperature below 10 °C (Figure 2a,b). Beside blowing cold air into the IJP, the Coolmobile 25 also sucked air from it, forming a closed thermal circuit between the IJP and the Coolmobile unit.

Three additional windows (transparent plates) were included in the IJP design: two on the downstream side for visual inspection of the IJP's interior during experiments; and another, with dimensions  $0.8 \times 0.18$  m<sup>2</sup>, at the bottom of the

185 upstream wall to feed PIV seeding particles into the IJP—in a closed volume—before starting an experiment. It was ensured that no leakage was present.

The ambient air in the testing chamber, without active control from heat exchangers, was around 25 °C. The frontier between IJP and testing chamber was a circular opening (nozzle) with a diameter D = 1 m, located at the center of the IJP's bottom panel. The nozzle was equipped with a honeycomb structure (Figure 2c)—a 100-mm thick hexagonal mesh with a grid

- diameter of 12 mm and an aluminium sheet thickness of 0.03 mm—designed to reduce the turbulence level and increase the homogeneity in the outgoing jet. The nozzle was positioned H = 3 m above the chamber floor, resulting in H/D > 1. For this configuration the confinement effects are negligible, allowing the primary vortex (PV) leading the downburst outflow to fully develop (Xu and Hangan, 2008; Junayed et al., 2019). A mechanism to control the nozzle's opening was developed to simulate the transient nature of real downbursts and to synchronize all the measurements from the start of the experiment.
- This was achieved with two rectangular wooden louvers held together by a central magnet (Figure 2d). At the desired time, the magnetic current was switched off, causing four elastic tensioners—two on each side—to pull the louvers apart instantly (time of complete opening about 0.3 s). Additionally, a piston with dimensions  $2 \times 2 \times 0.6$  m<sup>3</sup> was located at the top of the IJP (visible above the light blue walls of the IJP in Figure 2a,b) and was suspended from the testing chamber ceiling by means of a winch. A distance sensor measures piston speed and triggers the piston to stop before reaching the bottom of the
- IJP.

The IJP was mounted in the testing chamber on a robust wooden frame with planar dimensions of 5 (L)  $\times$  2 (W) m<sup>2</sup> (Figure 2a).
