# Peer review of "CLIMATHUNDERR: Experimental database of buoyancy-driven downbursts"

_Earth System Science Data, 2025_

## Author Comment (AC1)

The authors greatly thank the Reviewers and the Editorial Board for the work carried out. We did our best to take all the comments received into consideration, providing adequate answers to all of them.

All our answers and comments are reported in red color and two versions of the manuscript are provided. The first uses the word corrector to point out all our changes. The second accepts all these changes and corresponds to the final version of our new manuscript.

All line references below are referred to the revised version of the manuscript where changes are highlighted by the word corrector.

Reviewer #1 – Stefano Brusco

**General comments**

This manuscript presents an extensive and well-documented experimental database on downburst-like flows generated using buoyancy-driven and mechanically-assisted impinging jets in a large-scale thermal wind tunnel. The CLIMATHUNDERR project addresses an important, relevant and timely topic: the experimental replication of non-stationary thunderstorm downbursts under realistic thermodynamic conditions. The Authors propose a compelling contribution, succeeding in bridging a methodological gap between traditional gravity current and impinging jet techniques by reproducing buoyancy effects at large scales using only air as the working fluid. This work constitutes a notable advancement, and will influence the literature on this topic in the following years.

The experimental campaign is impressively ambitious, being definitely out of the current state-of-the-art. At the same time, the gathered database seems comprehensive. It combines advanced measurement techniques (large-scale PIV, distributed thermocouples), caused by a systematic variation of governing parameters (such as the variation of temperature, piston speed). It also includes the study of the problem on real-world orography. The processed datasets via Zenodo under open-access licensing will significantly enhances the paper's value for the community.

The manuscript is well-organized, clearly written, and follows a logical structure. The physical modeling choices, instrumentation, and experimental design are well justified.

However, some aspects would benefit from further clarification and elaboration, and I will be pointing them in the following. In fact, clarifying them will help contextualize the scope and limitations of the dataset for future users. In particular, these will concern the scaling of the experiments with respect to real-world phenomena, as well as the uncertainties in the data estimation.

Overall, the manuscript makes a valuable contribution to the wind engineering and atmospheric science communities. It meets the ESSD criteria for scientific quality, transparency, and data reusability. I recommend its acceptance after minor revisions, as outlined below.

We sincerely thank the Reviewer for the positive evaluation of our manuscript and for the careful and constructive review. We have addressed all the comments in detail and incorporated them into the revised version of the paper. In our view, the manuscript has been significantly improved thanks to the Reviewer's thoughtful and insightful suggestions.

**Specific Comments**

1. Section 3.3: several quantities withing Eqs. (1) and (2) are not introduced, unless I am mistaken. wB may be immediately introduced as the expected vertical buoyant jet velocities, avoiding detailing that w is the vertical velocity of the jet; concerning z, it is not specified whether it is pointing upwards or downwards, if I am not mistaken. The terms "t" is not introduced, and neither are the nomenclatures "d/dt", or the partial derivatives. Analogously, also several terms in Eq. (2) are not introduced. Besides, I would like to ask the Authors whether the hypothesis of non-varying reference density value could be verified.

We thank the Reviewer for this comment. All quantities are now properly introduced in the body text.

Regarding the reference density (1.23 kg m⁻³), its assumption as non-varying is justified because the temperature differences ($\Delta T$ up to 25°C) and pressure variations in the relatively small testing chamber (7 m height) are not large enough to cause significant density changes that would invalidate this approximation for the purpose of deriving the buoyant velocity component. The primary density difference that drives the flow is the initial $\Delta T$ between the IJP and the ambient air, which is accounted for in the buoyancy term. A sentence on this is added in the text.

2. Section 3.3: On the Richardson number, do the Authors think there is a variability in its estimation for full-scale downbursts? They refer to a specific case, but I am wondering whether it could vary across several full-scale records. Besides, I am wondering whether there is variability in the estimation of the Richardson number even in the experiments. Could the Authors provide more details about its variability with the other parameters of the tests ? I think it would make a very interesting point of the discussion.

This is a key point that directly addresses the variability of this crucial parameter, making the dataset even more accessible and valuable for users. The Richardson number ($Ri$), computed both theoretically and experimentally based on the total jet velocity (buoyancy + mechanical jet contributions), and the temperature difference $\Delta T$, is now reported in Table 2 for each experimental run. For the experimental

evaluation, $Ri$ is calculated using the vertical jet velocity $w$, defined as an average evaluated over the following dimensions:

1. Spatially, over the vertical and horizontal coordinate ranges 750-1250 mm and 0-250 mm, respectively;
2. Temporally, for the spatial region defined in (1), over the jet sub-interval during which $w$ can be regarded as quasi-constant. This time interval is identified as the longest continuous segment for which the temporal variation satisfies $dw < 5\%\, w_{\text{smooth}}$, where $w_{\text{smooth}}$ denotes the moving average of the spatially averaged velocity computed using a 0.5 s window.

The selection of these spatial and temporal ranges is motivated by the need to perform the calculation in a region where the jet velocity can be considered as stationary as possible. The slightly higher experimental values of $Ri$ compared with the theoretical estimates are attributed to the lower jet velocities observed in the experiments, resulting from flow dispersion and entrainment of the warmer ambient chamber air, which reduce the jet speed.

The Richardson number for full-scale downbursts indeed varies significantly. We calculated $Ri = \frac{g\,\beta\,\Delta T\,D}{w^2}$ for a representative full-scale downburst case with an equivalent base cloud jet diameter of 1000 m, temperature difference of 10 °C , and cloud-exit velocities of 10 to 20 m s⁻¹, yielding Ri values of approximately 3.6 and 0.9, respectively. These values reflect a transition from buoyancy-dominated to shear-dominated flow regimes, and we recognize that the downdraft velocity and temperature deficit are not independent; the former is largely driven by the integrated negative buoyancy. However, we acknowledge that Ri can vary significantly across different full-scale downburst events due to the different meteorological conditions driving these phenomena. For instance, downburst downdraft diameters may range from approximately 200 to 2000 m, temperature differences between the downdraft and ambient air may vary from roughly 5 to 15 °C or so, and outflow velocities can span 10 to >30 m s⁻¹, depending on storm intensity and environmental conditions (e.g., convective available potential energy or ambient wind shear). Using the Ri formula in the manuscript, these variations could lead to Ri values ranging from ~0.1 (high-velocity, shear-dominated flows) to ~10 (low- velocity, strongly buoyant flows). This variability, supported by analyses of full-scale downburst records (e.g., Fujita, 1985; Hjelmfelt,1988; Proctor, 1989), underscores that our single-case full-scale Ri estimate is a snapshot within a broader spectrum.

In our experiments, Ri also varied systematically. As noted, achieving very low jet velocities to perfectly match all full-scale Ri values is challenging due to stability requirements in the wind tunnel. It is important to acknowledge that, due to the significant geometric scaling D = 1 m in our experiments, the absolute Richardson numbers observed in our experiments vary between 0.01 and 0.10, and therefore are generally

lower than those typically found in natural downbursts. However, we designed the campaign to systematically vary ΔT (from 0°C to 25°C) and the mechanical jet velocity (0.8 m s⁻¹ and 1.3 m s⁻¹) across the test cases. Since Ri is a function of ΔT and the total downdraft velocity (w), each experimental run in Table 2 inherently possesses a unique Ri value. This experimental range of Ri allows to systematically investigate how the balance between buoyancy and inertial forces influences the flow dynamics under controlled conditions, providing valuable insights into trends and mechanism associated with varying Ri in downburst-like flows.

All this concept is summarized in Section 3.3.

3. Eq (5): It seems to me the Authors tried to estimate wIJ based on the flow rate conservation. Could this be instrumental to identify wGC (something that was discussed at Lines 244/245).

We thank the Reviewer for his pertinent comment. We evaluated $w_{IJ}$ as the mechanically-driven component of the jet, estimated from the piston's speed and nozzle's cross-sectional area (Eq. 5), which serves as a controlled mechanical input in our experiments.

$w_{GC}$ is **not** derived from $w_{IJ}$. Instead, our approach independently accounts for both components. $w_{GC}$ is theoretically calculated from the measured temperature differential (ΔT) using buoyancy principles (Eq. 3), as a direct thermal input. We have now explicitly included $w_B = w_{GC}$ in Table 2.

As reported in Section 3.3, the conceptual model for the predicted total jet velocity is the sum of these two independently determined components $w = w_{IJ} + w_{GC}$. This allows us to investigate the interplay of these distinct forces and how well the simple velocity superposition aligns with the PIV measurements, also reported in Table 2 as $w_{exp}$. $w_{GC}$ and $w_{IJ}$ are both inputs to the predicted total velocity that we then allow users to compare with the observed phenomena using the PIV data. A preliminary assessment of this is possible by comparing the theoretical and experimental values of $w$ and $Ri$ in Table 2.

4. Line 274: could the Authors provide details on how they synchronized all the different measurements based on the louvers' opening ?

Thank you. A paragraph is added at the end of Section 3.3.

5. Section 3.4.1: Do the simulated experimental phenomena "resemble" any of the full-scale events present in the database of the University of Genoa ? Specifically, the relative position of the orography model is allowing a touchdown point that may recall any full-scale event ? I believe comparison of wind velocity time-histories in specific points (i.e., anemometric stations in Genoa) will be a great point for future studies.

We thank the Reviewer for this insightful and relevant comment, with which we fully agree. The wind-monitoring network installed in the port of Genoa indeed provides records of full-scale downburst events that can be effectively compared with the experimental signals collected during our campaign. In particular, LiDAR scanner data offer valuable information on storm touchdown (especially for events starting over the sea) and on the associated flow dynamics. The selection of the orography model was also motivated by this objective, as it covers the same area—from the port of Genoa to the hilly region behind the city. Nevertheless, a detailed comparison between full-scale measurements and the present experimental data lies beyond the scope of this data manuscript and will be the subject of future dedicated research studies.

6. Do the Authors expect any end-effects induced by the boundary of the orography model that may cause some impact on the experimental measurements ? I am thinking how the orography model was installed within the surrounding environment in the laboratory. Was it flushing? I could not understand this piece of information from the manuscript. Perhaps an additional close-up picture would be beneficial.

Yes, the model's bottom surface (corresponding to sea level) was flush with the testing chamber floor. A sentence is added at line 304, while Figure 4 now includes two close-up pictures that better show the installation of the model in the chamber.

7. Section 3,5 and in general: How do the Authors handle measurement uncertainties of the relevant instrumentation ? Which is the impact that the tolerance of the instrumentation may have on the results ? I am thinking, for example, to the different parameters included in Table 2 for the same "nominal" experiment. Could measurement uncertainty be important in this discrepancy? Still on this topic, the Authors note (p. 18–19) that the exact reproduction of experimental conditions (e.g., ΔT, piston speed) was challenging, and that each test is to be considered "unique." While this is understandable, some discussion is needed on how such variability might affect data interpretation, particularly in cross-test comparisons or model training/validation. Are users encouraged to use specific test subsets (e.g., most stable ΔT cases)?

We thank the Reviewer for this relevant observation. Please find hereafter the measurement uncertainty of the experimental conditions presented in Table 2.

- Piston speed ($w_\mathrm{p}$) and jet speed ($w_\mathrm{IJ}$):

For the two target nominal jet speeds (0.8 m s$^{-1}$ and 1.3 m s$^{-1}$), the actual measured speeds are:

- 0.78 m s$^{-1}$ (average on 34 tests), with values ranging from 0.75 to 0.82 m/s;

- 1.30 m s$^{-1}$ (average on 23 tests), with values ranging from 1.26 to 1.35 m/s.

The measurement is based on the piston's position recorded just after the start and just before the end of its stroke, over a total displacement of 1.80 m. The DME 2000 laser distance sensor inside the IJP provides position data with a reproducibility error of ±2 mm every 29 ms. Based on this setup, the measurement uncertainties for $w_p$ and $w_{IJ}$ are estimated at approximately 0.8%.

The variations observed between tests are therefore not due to instrumentation uncertainty, but rather reflect the limits of mechanical system repeatability (winch speed accuracy and friction between the piston and the plenum walls).

Nevertheless, this repeatability remains satisfactory, especially when compared to the alternative of using a fan, for which it is particularly difficult to generate a slow, homogeneous airflow and to accurately monitor this low air velocity.

[Figure]

DME 2000 laser

• Temperature measurements:

The accuracy of type K thermocouples is approximately 1°C. This value includes sensor's accuracy and its repeatability. Therefore, when comparing temperature differences between tests, only repeatability errors contribute to the observed variations.

The temperature variations observed across experiments are mainly due to spatial and temporal inhomogeneities, such as the absence of temperature regulation inside the IJP, stratification in the testing chamber, and the thermal inertia of the floor and the IJP.

To compare or select the most consistent experimental configurations, stratification can be assessed using thermocouples $T_{WT,Tf19}$ and $T_{WT,Tf20}$, placed at 0 m and 2.90 m above the ground in the testing chamber, as well as $T_{IJP,Ts2}$ and $T_{IJP,Ts4}$, located at the bottom and top of the IJP.

[Figure]

Characteristics of type K thermocouples (https://omega.com)

8. Line 463- 466: as mentioned in the caption, the lines in Figure 9c is not a temperature timeseries, but it is the time-history of the percentage variation estimated from the thermocouples. I think this should be reflected in Line 465, replacing the terms "temperature timeseries" accordingly. This would also allow the Authors to modify the caption of Figure 9, avoiding the part "Temperature is actually the percentage …".

Thank you. The body text and caption of Figure 9 (now Figure 10) are modified accordingly.

9. Still on this Figure, I do not understand where are the surface thermocouples in Fig 9a-9b; could the Authors make them clearer ? Finally, I think that the legend voice "5.50 s" in Figure 9c may be replaced with "5.50 s + 0.5s".

Thank you, surface thermocouples are now included in Figure 9b (now Figure 10b) and the last legend item is updated.

10. The paper meets the general principles of open data, but it would benefit from a brief discussion of metadata structure, variable names, and standard formats used. Are NetCDF, CSV, or other structured formats adopted? Is the dataset interoperable with common post-processing tools?

Thank you. A paragraph discussing the metadata structure of the database is added at the end of Section 4.

*As final general conclusions, also to help with the next perspectives:*

11. It is well-known the significant effects played by the Reynolds number in numerous aspects related to Fluid Mechanics and aerodynamics of bluff-bodies, which affect the design of wind tunnel studies of structures for Wind Engineering purpose. I am wondering if the Authors expect analogous effects in this study concerning thermal aspects because of significant scaling effects studied here (1:2000 scale). I think it would be an interesting aspect to deepen in future studies.

This is a critical question for any scaled experimental study, especially one involving thermal effects. We fully acknowledge the extreme challenges associated with simultaneously matching all relevant dimensionless parameters (Reynolds number, Richardson number, geometric ratios like H/D) in wind tunnel experiments to full-scale phenomena. As is typical for wind tunnel studies of atmospheric flows, the experimental Reynolds numbers in our setup are significantly lower than those in full-scale downbursts. In our experiments, Re varies between $5.07 \times 10^4$ and $9.12 \times 10^4$ based solely on the mechanical velocity, and between $6.97 \times 10^4$ and $2.19 \times 10^5$ when the buoyancy-induced velocity component is also taken into account. A sentence is added in Section 3.3. While we try to achieve the highest possible Re within the facility's capabilities, perfect Re matching is absolutely not feasible at this geometric scale. We expect that this difference in Re will primarily influence the fine-scale turbulence characteristics of the flow. A new paragraph is also added in the Conclusions.

12. Still related to this aspect: In the introduction, it was mentioned that the models by Xhelaj et al. (2020, 2022, 2024) will be enhanced by incorporating the results from the current experiments to account for the thermal effects on downburst wind evolution. May I ask the Authors to detail how do they plan on doing that, also in light of scaling considerations ? What do they expect ?

We thank the Reviewer for this insightful comment. The model developed by Xhelaj et al. (2020, 2022, 2024) is currently 2D analytical/kinematic model that primarily describe the horizontal wind field at a generic height above the ground, focusing on mechanically-driven impinging jets. The CLIMATHUNDERR dataset will be fundamental in transforming and enhancing this model in several key ways:

1. Transition to a full 3D model with accurate vertical profiles: A primary enhancement will be the development of a full 3D model. This will involve an extensive study of the vertical/horizontal profiles of downburst outflow characteristics, which are critical for a full 3D representation. The CLIMATHUNDERR data, especially the LS-PIV velocity fields that capture both horizontal and vertical components across a significant height range, will allow us to accurately replicate these profiles of radial velocity component.

2. Deriving reduced-order models and temperature-dependent profiles: The detailed experimental data will enable advanced data analysis techniques. Specifically:

- Proper Orthogonal Decomposition (POD) analysis will be carried out on the PIV data to extract reduced-order models. These models will be important in identifying and accurately describing the dominant flow structures and the correct profile of the radial velocity, which is a current limitation of 2D model.

- The vortex structure, including its evolution and characteristics (e.g., core radius, height, strength), will be analyzed in detail from the PIV data.
- Crucially, the advanced analytical model will be developed to simulate the PIV data and will explicitly account for the temperature dependency of these profiles. This may be achieved by incorporating the Richardson number (Ri) as a governing parameter that quantifies the relative importance of buoyancy and inertial forces in the flow. This parameter, derived from the current experiments, will enable the analytical model to account for the thermal instability between the descending jet and the surrounding environment, thereby influencing the formulation of both the vertical velocity profile at peak intensity and possibly its temporal evolution.

  In terms of scaling, the model may be developed to preserve similar balance of key dimensionless groups—such as the Reynolds number (Re) and the Richardson number (Ri)—thus allowing the derivation of dimensionless relationships that can be transferred to full-scale scenarios.

The development of the enhanced 3D analytical model will fundamentally rely on dimensional analysis using the Buckingham Pi theorem. This approach is essential for constructing the model in a robust dimensionless form, which inherently serves to reduce the number of independent parameters required to describe the flow.

**Technical comments**

13. Line 246: is the term "wGC=" needed within the parenthesis, or "wB" should suffice ?

Thank you. "$w_{GC} =$" is now removed.

14. Line 259: I think there is a missing "m" after ground level( 0.10 … below the nozzle).

Thank you, corrected.

*Bibliography*

The adopted bibliography is adequate; however, I note the following potential issues:

15. Some journals are cited with their full name (e.g., Journal of Wind Engineering and Industrial Aerodynamics), others are cited with abbreviations. Concerning the latter, there are some that are cited with a final dot (e.g., Mon. Wea. Rev.) and others without it (e.g., Environ Fluid Mech). I suggest the Authors to be consistent throughout, following ESSD's guidelines.

Thank you, the references' format is now consistent throughout the bibliography.

16. Canepa et al. (2025) is not in the bibliography; could the Authors add it ?

Thank you, added.

17. In the text, it is always mentioned Canepa (2024), but in actuality in the bibliography there are two (a and b); I suggest the Authors to modify the text accordingly.

Thank you, the body text now reflects the actual bibliography.

Reviewer #2 – Anonymous

The authors describe a novel downburst experiment data set can be used for various types of studies, with particular interest for engineers.

The experimental data are valuable in that the experiments are the first to combine both of the widely used techniques for recreating downbursts – gravity current and impinging jet. However, the degree to which this is done successfully is difficult to determine, particularly since – as the authors say – each experiment in this study must be regarded as unique.

The organization of the paper and the writing are adequate, though there are some minor problems.

Overall, the authors do not adequately explain how downbursts develop and their observed spatial extent (in fact, what they do describe is incorrect). The authors also make reference to several types of vortices associated with downbursts but do not adequately describe these until late in the paper, and even then do not provide evidence that they exist in nature. This does not inspire confidence in the data or limited results that are presented.

The paper can, however, likely be improved with a number of minor revisions. Recommendations are described in the detailed comments below.

We sincerely thank the Reviewer for the time and effort dedicated to evaluating our manuscript. All comments have been carefully considered and addressed in detail, and the corresponding revisions have been incorporated into the updated version of the paper.

**Detailed Comments**

1. L14 – The abstract beings by discussing downbursts but on this line switches to the formation of thunderstorms. It took me a few reads to understand that "warm, humid air is lifted" does not refer to lifting by the horizontal outflow of a downburst. This needs to be revised to improve clarity for the reader.

Thank you. The sentence is now introduced by "The parent thunderstorm originates when…" (line 14).

2. L16 – This is the only line in the entire paper that tries to describe the origins of a downburst, and does so poorly. It should say that the air parcels sink due to 'negative buoyancy'. And it should be made clear – perhaps not in the abstract but at least in the introduction of the paper – that the negative buoyancy does not just happen on its own, that it requires evaporative cooling of precipitation.

We thank the Reviewer for this valuable comment. We agree that the original version lacked sufficient detail regarding the dynamics of downburst formation and development. A new paragraph has been added

in the Introduction, providing a more comprehensive description of the process, including the formation of individual vortical structures (see Q18) resulting from the thermodynamic instabilities that arise during downburst onset.

3. L17 – Re a 'symmetrical vortex', the authors should be clear about what that means i.e. symmetrical about what? And there isn't a need for an acronym e.g. 'PV' if that term is not used again in the abstract.

Thank you. The term "symmetrical" was in fact referring to "axisymmetric" and it has been removed, as symmetry is not always observed, as also demonstrated in our study. It has been replaced with "vortical structures" to emphasize that multiple vortices can form due to thermodynamic instabilities. In addition, the acronym "PV" has been removed from the abstract.

4. L45 – Re 'non-stationary winds', how can wind be non-stationary? Do the authors mean 'non-stationary wind phenomena', where the phenomena are non-stationary?

We thank the Reviewer for this observation. We agree that "non-stationary winds" may be misleading. The sentence has been revised to "non-stationary wind phenomena" to clarify that it refers to transient wind events, such as downbursts and gust fronts, whose statistical properties vary in time.

5. L46-47 – I don't believe that an increase in the frequency or intensity of such events has been shown, only projected. The authors need to revise this sentence to reflect that.

We thank the Reviewer for this clarification. We agree that current studies primarily project, rather than conclusively document, an increase in the frequency and intensity of such events. The sentence has been revised accordingly to reflect this.

6. L57 – The authors state the downbursts have a limited spatial extent of only a few km. That is incorrect. Downbursts can be tens of km across. If downbursts are 4 km or less, they are referred to as 'microbursts'. See Fujita (1981). If the authors are intending to investigate microbursts, and not downbursts in general, they should say so.

Thank you. The sentence is now corrected to "… over a limited spatial extent (a few to a few tens of kilometers) (Fujita, 1981)." (line 112).

7. L67 – Should be 'combined with thermal effects'

Thank you. Changed accordingly.

8. L72 – Should be 'a downburst develops'

Thank you. Changed accordingly.

9. L72 – Yes, a downburst is associated with relatively cold, dense air descending from aloft, but:

•        Needs to mention how i.e. evaporative cooling of precipitation

•        Needs to mention that it can be enhanced by precipitation loading

•        Needs to mention that 'dry downbursts' can occur in the absence of a thunderstorm, and how that can happen

We thank the Reviewer for the insightful comments. A new paragraph has been added in the Introduction implementing the three points above (see from line 61).

10. L93 – CAPE is not measured at the ground, it is measured through much of the depth of the troposphere. I'm sure at least some of the authors know this, so it's disappointing to see this here. It of course needs to be corrected.

We thank the Reviewer for this helpful clarification. We agree that Convective Available Potential Energy (CAPE) is not measured at the ground, but rather represents the vertically integrated buoyant energy available to an air parcel as it rises through the troposphere. The sentence has been revised to reflect this more accurately.

11. L107 – The term 'wind-safer' is not valid; the authors need to find another way to express this.

We thank the Reviewer for this observation. The term "wind-safer" is maintained as it appears in the official title of the research project THUNDERR.

12. L117 – What is the difference between 'meteorological' and 'anemometric' measurements? I think just meteorological measurements would suffice here.

We thank the Reviewer for this thoughtful comment. We agree that meteorological measurements would generally be sufficient; however, in this context, we aimed to distinguish between conventional meteorological observations (typically collected at lower sampling frequencies) and high-frequency anemometric measurements obtained from ultrasonic anemometers or LiDARs. To clarify this distinction, we have revised the sentence accordingly.

13. L119-120 – I think the Mediterranean Seas serving a source of warn and humid air for thunderstorm updrafts is an assumption made by the authors. If not, please provide references.

Thank you. References are now provided (line 185).

14. Figure 1 – There is a dotted white line on the upper plenum – is there supposed to be a vertical measurement shown here?

Thank you. The vertical dotted line in the IJP now shows the corresponding measurement.

15. Figure 1 – The yellow arrow appears to show only one orientation, so not sure why 'upstream/downstream orientations' is mentioned in the caption.

Thank you. The caption now reports only "upstream orientation" referred to the yellow arrow in the figure.

16. L177 – 'in the view of ensuring' should be 'in order to ensure'

Thank you. The sentence is now changed accordingly.

17. L190 – Is the sheet really only 0.03 mm thick? Just want to make sure this isn't a typo.

Thank you. We contacted the supplier, who informed us that the actual thickness is 0.07 mm, as indicated by the reference "2.6 1/2 .003 3003 T=100MM 1250 x 2500 MM", which specifies a thickness of 0.003 inch. The text is changed accordingly.

18. L192 – The authors mention a primary vortex, a secondary vortex, and trailing vortices, but these are not described in detail and for the latter two the origins are not explained. There needs to be a paragraph or two at this point in the manuscript that describes these and refers to a idealized schematic that shows all of the vortices occurring at once (based on Fig 8c). Showing what these vortices are so late in the paper (Fig. 8) is not adequate. The authors also need to be state whether the SV and TV have ever actually been observed to occur.

We thank the Reviewer for the comment. A paragraph describing the formation and evolution of the individual vortical structures has been added in the Introduction (see response to Q2), together with a schematic figure (see Figure 1).

19. L222 – Not sure if 'an overpressure' is an actual engineering term – if not, please revise.

Thank you. "An overpressure" was replaced by only "overpressure" which is a technical term in engineering or fluid mechanics contexts.

20. L311 – Again, the PV and SV need to be explained earlier with reference to an idealized schematic.

See response to Q2 and Q18.

21. L317 – 'allows to extend' should be 'allows for'

Thank you. The sentence is changed accordingly.

22. Figure 4. Spectacular results here. The caption mentions 'vortices' but should say whether they are PV, SV or TV or some combination.

Thank you. "Vortices" is now replaced by "PV".

23. Figure 4. The labels on the axis for b are too small to be legible. Need to increase the font size.

Thank you. The axis labels are now removed, as they are not relevant to the purpose of the figure and their removal improves its clarity and visual balance.

24. Figure 5. Caption should say what the arrows and the broken line mean and that the green rectangle represents the PIV field of view.

Thank you. The caption of Figure 5 is now updated to integrate these details.

25. L350 – Should be '0.5 mm' and '300 mm'

Thank you, modified.

26. Figure 6. In caption, 'model' should be 'orography model' for clarity.

Thank you, modified.

27. L391 - 'model' should be 'orography model' here for clarity

Thank you, modified.

28. L432 – Re '1000 ms', why not just write '1 s'?

Thank you for the suggestion. We have revised the text to preserve both the clarity and the precise timing. We chose to write 1000 ms instead of 1 s to reflect the level of precision involved in the synchronization process. The synchronization system had a resolution of 1 ms, and the power-on time of the winch using a Solid-State Relay was 10 ms. Using only 1 s could suggest a lower precision than what was achieved.

29. Figs 8 and 9 – It's difficult to make out the axis titles – please correct.

Corrected, thank you.

30. Figure 9. Re 'Temperature is actually', should probably say 'the y-axis is' – it's not temperature.

The caption is already modified according to Q8 from Reviewer #1.

31. L483 – Where would these 'full-scale recordings of downburst events' come from? The authors should expand on this in the text.

Thank you. The paragraph is now extended to specify the sources of full-scale downburst recordings.

32. References – a paper on downbursts that does not reference Fujita – who discovered downbursts – is incomplete.

Reference

Fujita, 1981: Tornadoes and downbursts in the context of generalized planetary scales. J. Atmos. Sci., 38, 1511-1534.

Thank you. We agree with the Reviewer and the suggested reference is now added in the paper.